# Super-resolution architecture of mammalian centriole distal appendages reveals distinct blade and matrix functional components

T. Tony Yang[1], Weng Man Chong[1], Won-Jing Wang[2], Gregory Mazo[3], Barbara Tanos[4], Zhengmin Chen[1], Thi Minh Nguyet Tran[1,7], Yi-De Chen[1], Rueyhung Roc Weng[1,8], Chia-En Huang[1,9], Wann-Neng Jane[5], Meng-Fu Bryan Tsou[3] & Jung-Chi Liao [1,6]

Distal appendages (DAPs) are nanoscale, pinwheel-like structures protruding from the distal end of the centriole that mediate membrane docking during ciliogenesis, marking the cilia base around the ciliary gate. Here we determine a super-resolved multiplex of 16 centriole-distal-end components. Surprisingly, rather than pinwheels, intact DAPs exhibit a cone-shaped architecture with components filling the space between each pinwheel blade, a new structural element we term the distal appendage matrix (DAM). Specifically, CEP83, CEP89, SCLT1, and CEP164 form the backbone of pinwheel blades, with CEP83 confined at the root and CEP164 extending to the tip near the membrane-docking site. By contrast, FBF1 marks the distal end of the DAM near the ciliary membrane. Strikingly, unlike CEP164, which is essential for ciliogenesis, FBF1 is required for ciliary gating of transmembrane proteins, revealing DAPs as an essential component of the ciliary gate. Our findings redefine both the structure and function of DAPs.

[1] Institute of Atomic and Molecular Sciences, Academia Sinica, Taipei 10617, Taiwan. [2] Institute of Biochemistry and Molecular Biology, National Yang Ming University, Taipei 11221, Taiwan. [3] Cell Biology Program, Memorial Sloan-Kettering Cancer Center, New York, NY 10065, USA. [4] Institute of Cancer Research, London SW7 3RP, UK. [5] Institute of Plant and Microbial Biology, Academia Sinica, Taipei 11529, Taiwan. [6] Genome and Systems Biology Degree Program, National Taiwan University, Taipei 10617, Taiwan. [7] Present address: Institute of Biotechnology, Vietnam Academy of Science and Technology, Hanoi, Vietnam. [8] Present address: Department of Internal Medicine, National Taiwan University Hospital, Taipei, Taiwan. [9] Present address: TFBS Bioscience, Inc., Taipei, Taiwan. Correspondence and requests for materials should be addressed to M.-F.T. (email: tsoum@mskcc.org) or to J.-C.L. (email: jcliao@iams.sinica.edu.tw)

All cilia grow from the distal end of centrioles/basal bodies[1–4]. To initiate primary ciliogenesis, a mature vertebrate centriole docks to the membrane, recruits membrane vesicles, regulates axoneme growth, and forms part of the ciliary base that gates the ciliary compartment, all of which depend on its distal end[5–9]. In particular, distal appendages (DAPs) are previously defined as nine-bladed, pinwheel-like structures protruding from the distal periphery of vertebrate centrioles[10] and are thought to be involved in many of the aforementioned steps of ciliogenesis. CEP83, CEP89, SCLT1, CEP164, and FBF1 are core DAP components recruited to centriole distal ends before and independently of ciliogenesis[5]. These components are assembled at the G2/M border and assembly is dependent on C2CD3, another centriole-distal-end protein[11]. In response to cell cycle cues in G1 that stimulate ciliogenesis, other proteins including TTBK2, recycling endosome components (Rab8, Rab11, and EHD1), IFT88, and ARL13B are recruited to DAPs along with the associated membrane vesicles[12–17]. Upon membrane docking mediated by DAPs, centriole distal ends are further modified to release negative regulators of ciliogenesis, such as CP110 or CEP97[18], which enables nucleation of the axoneme from the centriole. At the nascent axoneme bud, a specialized structure known as the transition zone (TZ) is built as part of the ciliary barrier from dozens of protein components recruited from the cytoplasm[19], guided in part by the residential centriole-distal-end proteins CEP290 and CEP162[20,21]. The continuous growth and maintenance of the axoneme in the gated ciliary compartment are further supported by the intraflagellar transport (IFT) machinery[22], generating a functionally intact primary cilium that serves as a sensory organelle.

By initiating ciliogenesis, DAPs are situated in a highly intricate multi-junction position that marks the border between the centriole and axoneme (or TZ), as well as the junction between the plasma membrane and the ciliary membrane[5–9]. This unique location raises the possibility that DAPs and their associated factors may also form part of the structural barrier gating the ciliary compartment. However, the pinwheel-like morphology, as defined by electron microscopy (EM), does not intuitively suggest how DAPs can serve as a barrier, given that the large space between each pinwheel blade would allow many substances to pass through with little or no resistance. Thus, DAPs may simply function as the anchor mediating membrane docking or IFT machinery recruitment. Alternatively, however, the pinwheel shape may not morphologically reflect the full complexity of DAPs, due the limitations of EM.

Here we resolve 12 DAP-associated molecular species and four additional proteins surrounding DAPs using direct stochastic optical reconstruction microscopy (dSTORM)[23,24], revealing a three-dimensional (3D) nanoscopic resolution of the protein megacomplex in intact cells and reaching a level of detail not previously achieved by current structure biology techniques. We find the architecture supports a previously unrecognized structure that gates transmembrane proteins. Together, our findings reveal an unprecedented architectural and functional framework at the base of mammalian primary cilia.

## Results

**dSTORM reveals distinct radial occupancies of DAP proteins.** To systematically characterize DAPs and centriole distal ends, our dSTORM imaging analysis focused on core DAP components (CEP83, CEP89, SCLT1, CEP164, and FBF1), DAP-associated proteins (Cby1, TTBK2, and IFT88), other centriole-distal-end proteins (C2CD3, CP110, CEP97, CEP162, and CEP290), ciliary membrane proteins (ARL13B and EHD1), and the subdistal appendage (sDAP) component ODF2. Super-resolution microscopy has been used previously to localize centriolar proteins[25–29], but systematic determination of proteins at the centriole distal end has never been performed. Here, mature centrioles oriented axially or laterally (perpendicular or parallel to the imaging plane) in human retinal pigment epithelial (RPE-1) cells were imaged (estimation of localization precision shown in Supplementary Fig. 1). As depicted in Fig. 1a, visualization of axially and laterally oriented centrioles, respectively, allows clear determination of the

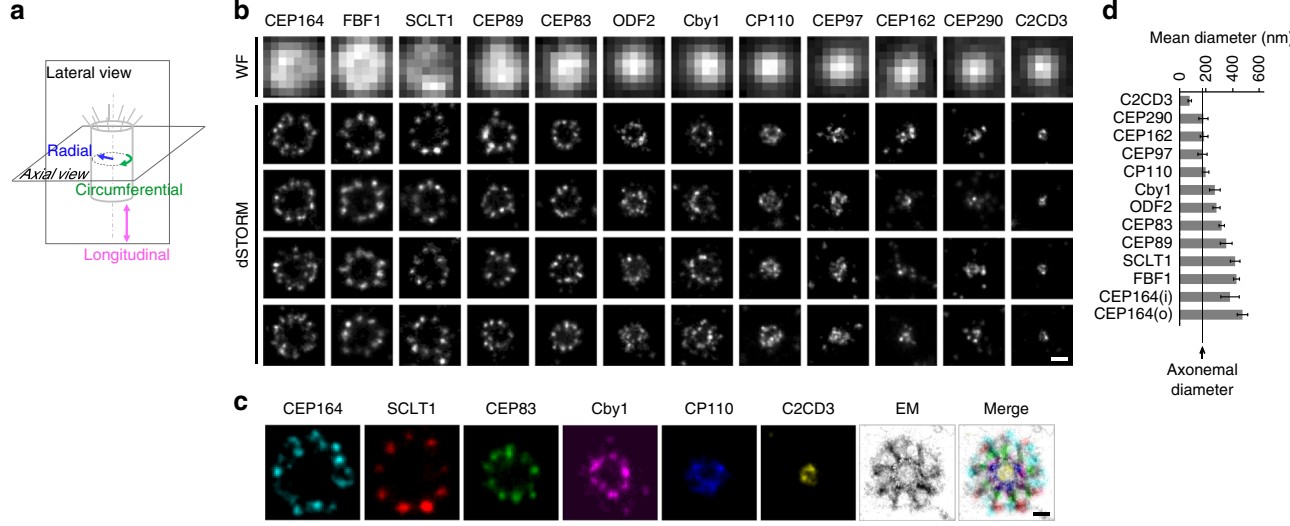

**Fig. 1** Radial localization of proteins at the distal appendage (DAP) region. **a** Definitions of view angles (axial and lateral) and orientations of molecular distributions (radial, circumferential, and longitudinal) on a mother centriole. **b** dSTORM super-resolution images showing the distribution patterns of various DAP, sub-distal appendage, and centriolar proteins in order of radial size, which are not resolvable by wide-field (WF) imaging. **c** Axial view EM image overlaid with single-color super-resolution images of CEP164, SCLT1, CEP83, Cby1, CP110, and C2CD3, illustrating that CP110 overlaps with axonemal microtubules, whereas SCLT1 and CEP164 are localized near the tips of highly electron-dense branches of the EM image that represent DAPs. **d** Mean diameter analysis revealing dimensional differences among ciliary proteins in **b**, where the diameters of CP110, CEP97, CEP162, and CEP290 are similar to that of the axoneme measured by EM (dashed line). CEP164 (o) and CEP164 (i) represent the outer and inner diameters of CEP164, respectively. The results (presented as mean ± SD) and the numbers of measured points are summarized in Supplementary Table 1. Bars **b** = 200 nm and **c** = 100 nm

radial and longitudinal distributions of centriolar components, leading to a 3D architectural map of the centriole distal end.

Analysis of axially oriented centrioles revealed radial localization of core DAP components exhibiting nine-fold symmetric ring-like patterns that differ in size/diameter, with CEP164 and FBF1 forming the largest ring, followed by SCLT1, CEP89, and CEP83 (Fig. 1b and Supplementary Table 1). CEP164 displays a broader radial distribution with distinct outer and inner intensity peaks (Fig. 1b and Supplementary Fig. 2), resembling the pinwheel-like pattern characteristic of DAPs as defined by EM (Fig. 1c). By contrast, C2CD3, a protein essential for DAP assembly, occupies a surprisingly small, more compact region inside the centriole lumen than CP110, CEP97, CEP290, and CEP162 (Fig. 1b, c), all of which form a ring-like distribution of ~ 170−200 nm in diameter (Fig. 1d, Supplementary Fig. 3, and Supplementary Table 1), similar in size to the centriolar axoneme (Fig. 1c). This suggests that C2CD3 is not part of the DAP slanted structure defined by EM; rather, it is a luminal protein likely defining the distal environment of centrioles required for appendage formation.

**FBF1 localization differs from those of other DAP proteins.** Two-color dSTORM imaging of the radial distribution further revealed distinct angular positions among the core DAP proteins (Fig. 2a-e). Images of pairs of CEP83-SCLT1 (Fig. 2a), CEP89-CEP164 (Fig. 2b), and CEP164-SCLT1 (Fig. 2c) showed that they can all be allocated to tilted lines resembling the arrangement of

the nine electron-dense DAP blades observed in EM images. SCLT1 was often localized within the blades of CEP164 propeller-like signals (Fig. 2c and Supplementary Fig. 4). By contrast, FBF1 fills the circumferential gaps of CEP164 puncta (Fig. 2d) as well as SCLT1 puncta (Fig. 2e), likely localized to an open area between adjacent electron-dense DAPs observed by EM. Histograms of angular spacing between two puncta, one from each of a pair of DAP proteins collected from multiple cilia, revealed that CEP83-SCLT1, CEP89-CEP164, and CEP164-SCLT1 pairs yielded peaks close to multiples of 40°, or 360°/9 (Fig. 2f, Supplementary Fig. 5, and Supplementary Table 2), reflecting their similar circumferential arrangements. By contrast, histograms for CEP164-FBF1 and FBF1-SCLT1 pairs revealed peaks close to 20°, 60°, etc. (Fig. 2f and Supplementary Table 2), confirming their alternating arrangements. Thus, unlike other core DAP components, which localize to the slanted DAP blades (Fig. 2a−c, g), FBF1 is instead associated with the space between DAP blades (Fig. 2d, e, g).

**Axial and lateral images reveal a 3D molecular map of DAPs.** The lateral view, revealed by a series of two-color dSTORM images (Fig. 3a), showed that CEP83, SCLT1, and FBF1, respectively, form distinct thin layers along the proximal-to-distal axis of the centriole (Fig. 3b). Consistent with its wide radial distribution, CEP164 also occupies a broad longitudinal localization, forming a trapezoid-like shape with two tapered sides, resembling that of the DAP slanted structure observed by EM (Fig. 3b, d). This suggests that CEP164 forms the primary backbone of DAPs.

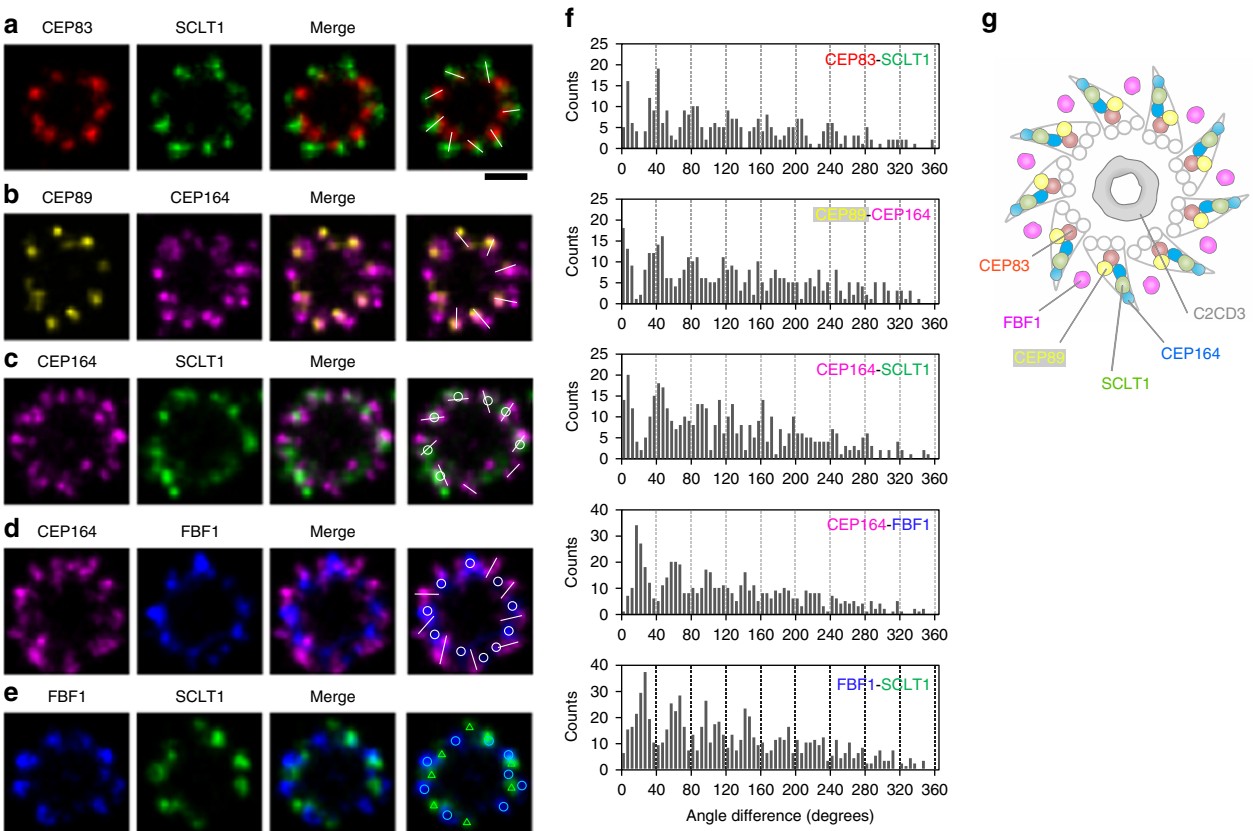

**Fig. 2** Angular distributions of CEP83, CEP89, SCLT1, FBF1, and CEP164. **a–e** Two-color axial-view images of **a** CEP83-SCLT1, **b** CEP89-CEP164, **c** CEP164-SCLT1, **d** CEP164-FBF1, and **e** FBF1-SCLT1 pairs. CEP83 and SCLT1 **a**, and CEP89 and CEP164 **b** align well with each other. SCLT1 likely resides within each "blade-like" distribution of CEP164 signals **c**, whereas the angular positions of FBF1 puncta are different from those of CEP164 puncta **d** and SCLT1 puncta **e**. White bars mark SCLT1 in **a** or CEP164 in **b–e**; circles mark SCLT1 in **c** or FBF1 in **d–e**; triangles mark SCLT1 in **e**. **f** Angular analysis in the circumferential direction shows that CEP83, CEP89, SCLT1, and CEP164 exhibit similar angular distributions, while FBF1 is arranged alternately with SCLT1 or CEP164, suggesting it fills the gaps between adjacent CEP164 puncta. **g** Model of the localization of DAP proteins. FBF1 occupies the gaps between DAP blades, whereas C2CD3 localizes to the centriolar lumen. Bar **a–e** = 200 nm

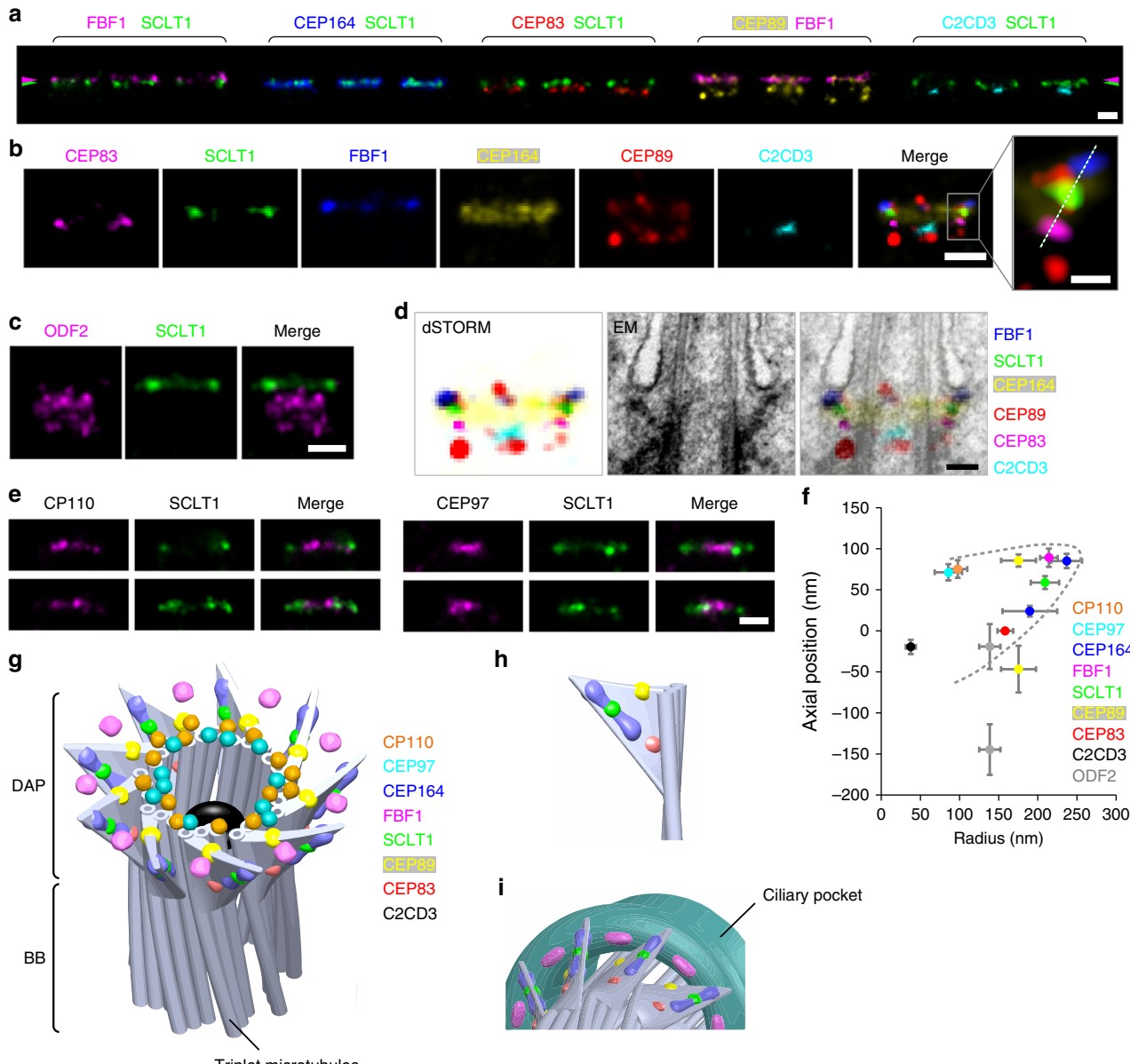

**Fig. 3** Three-dimensional spatial map of distal appendage (DAP)-associated proteins. **a** Three representative two-color dSTORM images from a lateral view of each pair of DAP proteins illustrates their relative longitudinal positions. Pink and green arrowheads depict the heights of FBF1 and SCLT1, respectively. **b** Representative images of DAP proteins are aligned and combined according to their average longitudinal positions. The magnified image (inset) demonstrates the mapping of DAP proteins on the tilted arrangement of the DAP structure (dashed line). **c** Super-resolved images of the subdistal appendage protein ODF2 reveal its two-layered localization at a far lower location than that of SCLT1. **d** Merged super-resolution image in **b** overlaid with a longitudinally sliced EM micrograph. CEP83, SCLT1, and FBF1 are well aligned with the electron density of the slanted DAPs. **e** CP110 and CEP97 are localized longitudinally, similar to SCLT1, but with a narrower lateral width. **f** Combined relative localization of DAP-associated proteins in radial and longitudinal directions, revealing the envelope of the slanted arrangement of a DAP blade (dotted line). The results (presented as mean ± SD) and the numbers of measured points are summarized in Supplementary Table 1. **g** A 3D computational model illustrating the positioning of proteins at the DAP region with a backbone structure based on EM[10]. **h** CEP164, CEP83, SCLT1, and the distal layer of CEP89 mapped onto a blade of the 3D-reconstructed DAPs. **i** FBF1 localizes to gaps between two adjacent DAP blades, possibly at the proximal boundary of the ciliary pocket. Bars **a**, **b** (left), **c**, and **e** = 200 nm; **b** (inset) = 50 nm

Meanwhile, CEP89 occupies two layers at the centriole distal end both with a similar width but separated axially by ~ 130 nm (Fig. 3b and Supplementary Fig. 6). The distal layer of CEP89 is close to the same longitudinal position as FBF1 (Fig. 3b and Supplementary Fig. 7), whereas the proximal layer is located longitudinally close to ODF2 in sDAPs (Fig. 3c and Supplementary Fig. 7), suggesting possible dual roles for CEP89 in DAPs

and sDAPs. By overlaying the dSTORM signals over EM images of ciliated centrioles, we found that CEP164, CEP83, SCLT1, and FBF1 align well with DAPs, with FBF1 localizing closest to the ciliary membrane (Fig. 3d). CP110 and CEP97 in non-ciliated centrioles are localized at a longitudinal position, similar to SCLT1 (Fig. 3e). Collectively, the relative positions of DAP proteins in the lateral view reflect the contour of a DAP blade

(Fig. 3f) where the diverse localization variations among different proteins suggest their largely different distribution ranges in space.

Combining information from all relative axial and lateral positions, we built a 3D computational model to visualize the positioning of proteins at the DAP region (Fig. 3g), with a backbone structure illustrated based on EM[10]. CEP164, CEP83, SCLT1, and the distal layer of CEP89 mapped well onto the 3D-reconstructed DAPs (Fig. 3h). By contrast, FBF1 localizes to the distal regions of the DAP gaps between adjacent CEP164 molecules, possibly near or on the proximal boundary of the ciliary pocket (Fig. 3i). Thus, FBF1 may serve a role distinct from other core DAP components.

**ARL13B lies adjacent to DAPs during ciliary initiation.** Building upon this 3D nanoscale map of DAPs, we continued to add DAP-associated proteins to the map. We first examined the spatial relationship between DAPs and the membrane. ARL13B is a small GTPase associated with the ciliary membrane and ciliary vesicles[30,31]. During the early stages of cilia growth (following an 8 h serum starvation), ARL13B puncta were localized in close proximity to FBF1 puncta (Fig. 4a, b). However, after cilia had grown fully (after a 24 h starvation), ARL13B signals occupied the entire primary cilium, similar to IFT88 signals (Supplementary Fig. 8), but were completely excluded from the ciliary base above

DAPs where the proximal TZ was located (Fig. 4c and Supplementary Fig. 9). These results suggest that ciliary vesicles carrying ARL13B may initially dock at the DAP region, but are later excluded from at least the proximal end of the TZ of intact cilia, consistent with the previous observation of a ciliary zone of exclusion[32]. EHD1, a protein involved in ciliary vesicle formation, was found to occupy a space wider than the diameter of the primary cilium (Fig. 4d, Supplementary Fig. 9, and 10; Supplementary Movie 1), consistent with its presence on the ciliary pocket membrane[15].

We also examined the localization of TTBK2, a kinase interacting with CEP164 and promoting ciliogenesis[12,33]. In proliferating cells, TTBK2 was sporadically scattered with an asymmetric and incomplete coverage around the centriole (Fig. 4e). However, upon serum starvation, a broadened radial and longitudinal distribution of TTBK2 was induced, forming a thicker occupancy covering the distal regions of DAPs close to SCLT1 (Fig. 4e−g and Supplementary Fig. 11).

**IFT88 and FBF1 localize to the DAP matrix.** We next examined IFT88, a core IFT complex component known to be associated with DAPs[25]. We found that IFT88 occupied a broad angular distribution without clearly distinguishable nine puncta in the axial view (Fig. 5a). Lateral views revealed that in addition to being distributed along the entire ciliary axoneme, IFT88 is

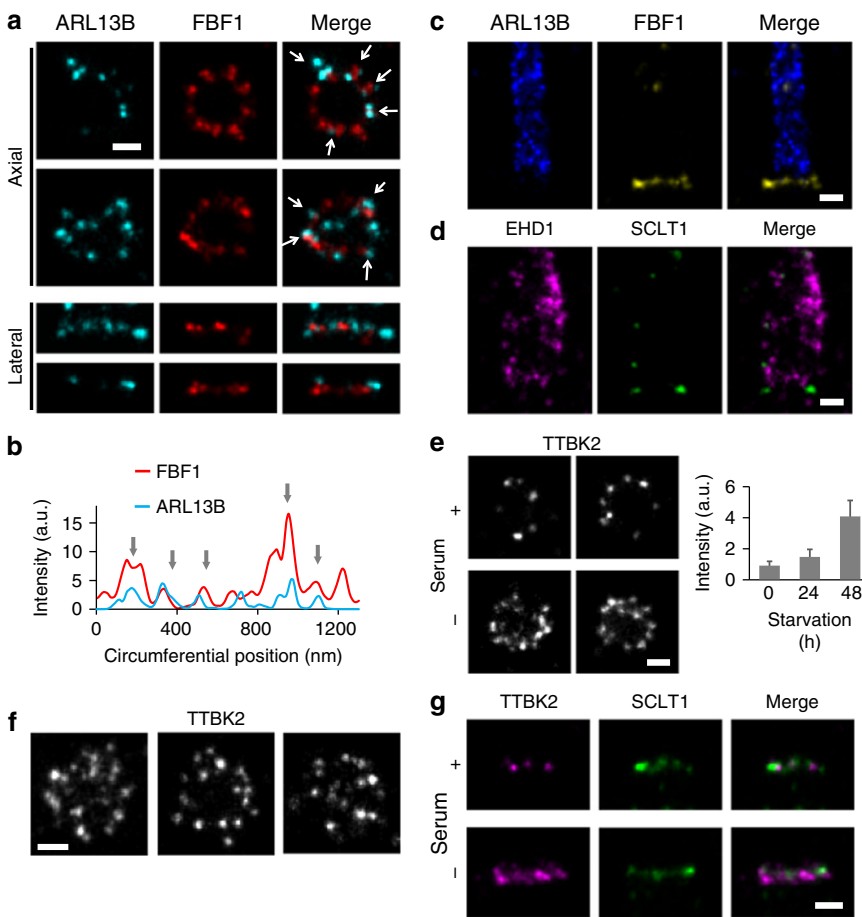

**Fig. 4** Localization of other DAP-related proteins ARL13B, EHD1, and TTBK2. **a** During the early stages of cilia growth, ARL13B is located adjacent to FBF1 (arrows), as confirmed by correlation analysis in **b**. **c** ARL13B becomes localized to cilia during or after ciliogenesis with an exclusion zone approximately at the proximal transition zone. **d** The vesicle-forming protein EHD1 is localized to a peripheral region wider than that of ARL13B, presumably on the membrane of the ciliary pocket. Axial **e**, **f** and lateral **g** views of the distribution of TTBK2 with and without serum, showing a broader and more enriched TTBK2 distribution in both radial and longitudinal directions after serum starvation. Results are presented as mean ± SD of 24, 20, and 23 centrioles for 0, 24, and 48 h, respectively. Bars = 200 nm

heavily concentrated at the ciliary base, at a longitudinal position spanning and proximal to the SCLT1 molecules (Fig. 5b, c and Supplementary Fig. 12). To further uncover the exact localization of IFT88 close to DAPs, we performed 3D two-color dSTORM imaging of IFT88 and SCLT1. We found that IFT88 occupied a conical distribution (Fig. 5d, e, Supplementary Fig. 13, and

Supplementary Movie 2), resembling the tapered shape of DAPs. Strikingly, the relative distributions of IFT88 and SCLT1 reveal abundant IFT88 proteins filling the gaps between neighboring SCLT1 molecules (Fig. 5f), demonstrating that molecules can occupy or be retained in this unexplored area between DAP blades. We therefore designated the regions between the nine DAP blades

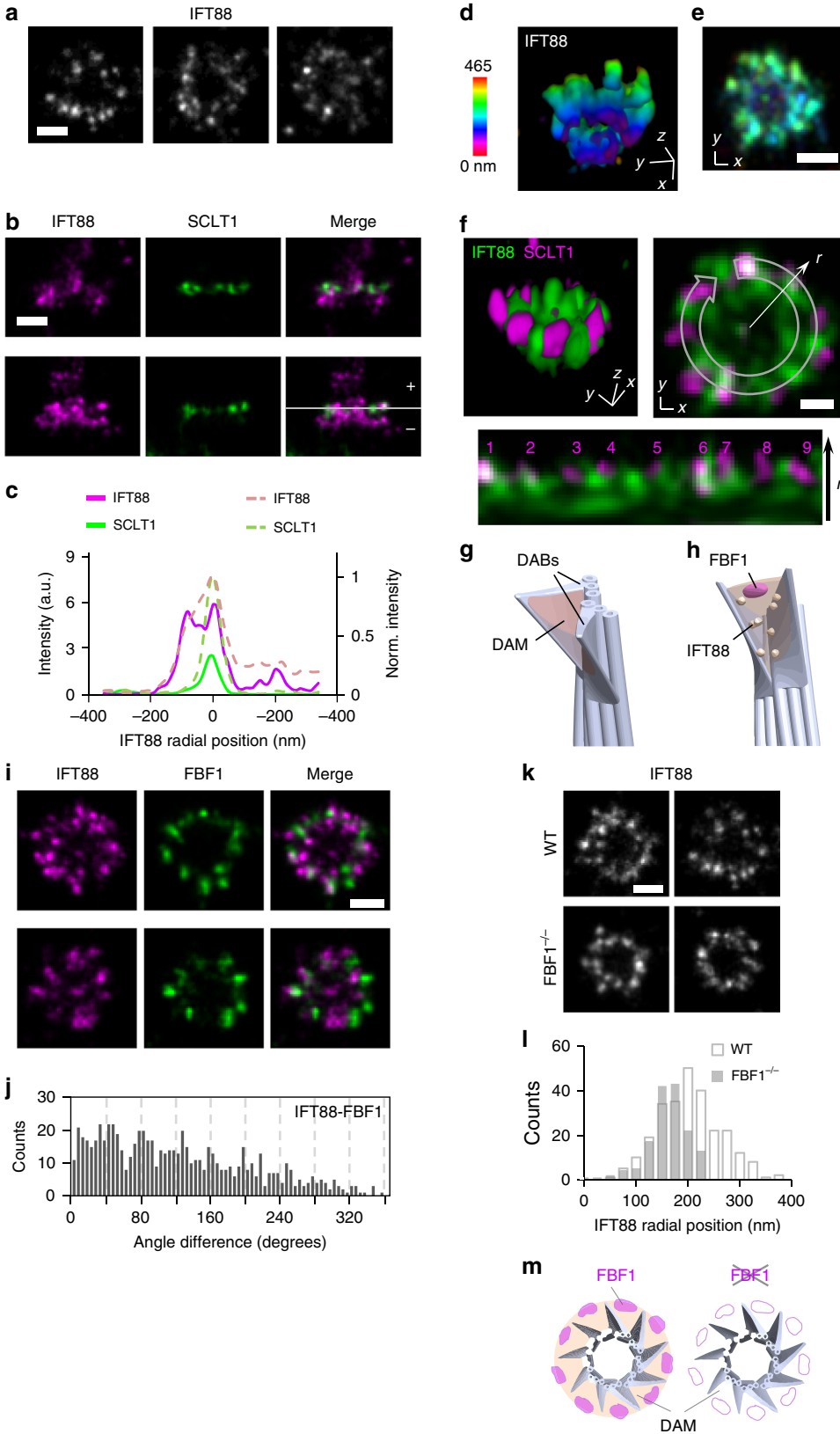

as the DAP matrix (DAM), in which proteins such as IFT can be concentrated. In this sense, the intact DAP structure should be viewed as a conical addition at the centriole distal end consisting of the distal appendage blades (DABs) and the DAM (Fig. 5g).

The positioning of FBF1 at DAP gaps toward the ciliary membrane suggests that FBF1 localizes at the distal end of the DAM interface with the ciliary membrane (Fig. 5h). Indeed, two-color IFT88-FBF1 dSTORM images showed that IFT88 coexists at the angular positions of FBF1 (Fig. 5i) and histograms of angular spacing revealed peaks close to multiples of 40° (Fig. 5j), confirming the positioning of FBF1 at the DAM. To check whether FBF1 and IFT88 at the DAM are functionally related, we generated the $FBF1^{-/-}$ CRISPR knockout line (Supplementary Fig. 14) and examined IFT88 localization in this region. Notably, in $FBF1^{-/-}$ cells, the distribution of IFT88 was reduced to a size comparable with that of CEP83 (Fig. 5k, l), suggesting that IFT88 cannot span the DAM region without FBF1. The causal effect of FBF1 depletion on IFT88 distribution is consistent with the

observation of interactions between FBF1 and IFT88 in a previous study[34]. Our results potentially imply a functional role for FBF1 in maintaining the DAM spatial arrangement (Fig. 5m).

We next tested whether DAB components perform a role in the structural arrangement of the DAM, and vice versa. Axial-view dSTORM imaging revealed no obvious change in CEP164 distribution upon FBF1 knockout (Fig. 6a, c). By contrast, knockdown of CEP164 did reduce the radial distribution of FBF1 (Fig. 6b, d). Thus, without CEP164 at the DAB tips, even though FBF1 was recruited to the DAP region[5], it was unable to reach the outermost DAM location (Fig. 6e). CEP164 depletion has been shown to affect IFT recruitment[35] and it is also possible that the DAM was damaged upon knockdown of CEP164, resulting in the collapse of the FBF1 ring.

Subsequently, we expanded our 3D computational model to include all ciliary structures examined, including the membrane, pocket, axoneme, DABs, and the DAM (Fig. 6f, g, Supplementary Fig. 15, and Supplementary Movie 3).

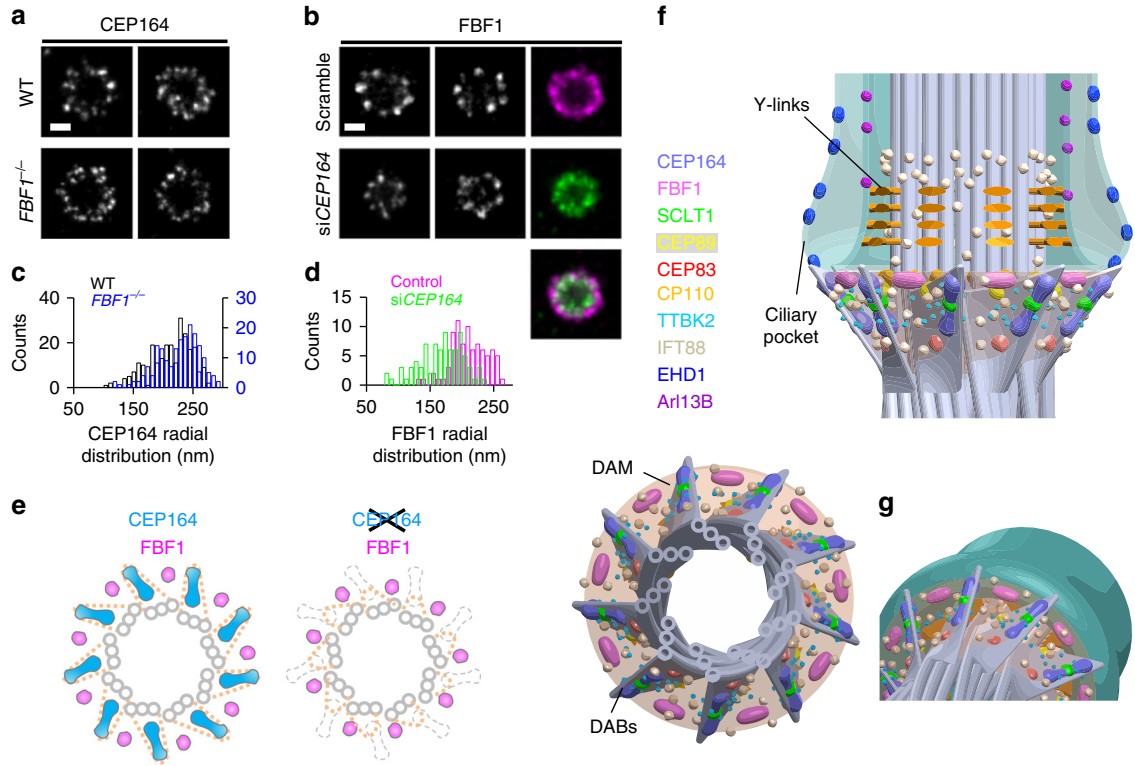

**Fig. 6** DAB components mediating the structural arrangement of the DAM. **a**, **b** Axial view dSTORM images reveal the distribution of **a** CEP164 and **b** FBF1 at the mother centrioles upon CRISPR/Cas9 *FBF1* knockout and *CEP164* siRNA silencing, respectively. It is noteworthy that the FBF1 ring formed upon *CEP164* knockdown becomes smaller than that of controlled cells (**b**, right panel). **c**, **d** Statistical analysis of the radial distribution of CEP164 and FBF1 upon perturbation. The size of CEP164 in $FBF1^{-/-}$ cells is comparable to that in WT cells **c**, whereas the size of FBF1 is reduced upon *CEP164* knockdown **d**. **e** Model showing that the arrangement of FBF1 is no longer maintained at the outer region of the DAM in the absence of CEP164. **f** Three-dimensional model based on all axial and lateral super-resolution images illustrating the molecular architecture on the framework of DABs, DAM, axoneme, and ciliary membrane. **g** Close-up view of the DAP region, suggesting that FBF1 potentially interfaces the ciliary membrane and the DAM. Bars = 200 nm

**Fig. 5** Localization of IFT88 revealing a distal appendage matrix (DAM). **a** Three representative axial-view dSTORM images of IFT88 reveal a broad angular distribution without a clear nine-fold symmetric pattern. **b** Two representative sets of two-color images showing IFT88 spanning a longitudinal range relative to the location of SCLT1 (marked with a dashed line). The solid line in **c** depicts the distribution in the lower panel in **b**, whereas the dotted line illustrates normalized distributions averaged over nine data pairs. **d**, **e** 3D STORM imaging of IFT88 showing the tapered distribution of IFT88 that is wider at the distal end of the DAP region. **f** Three-dimensional two-color super-resolution image revealing IFT88 proteins in the gaps between neighboring SCLT1 regions, as shown in the lower image (derived from the upper-right image) in which nine SCLT1 puncta are numbered accordingly. **g** The region between adjacent DABs is designated as the DAM. **h** Model showing the localization of FBF1 and IFT88 at the DAM. **i** Two-color imaging of IFT88-FBF1 shows the angular coincidence between IFT88 and FBF1 at the DAM. **j** Histogram of angular spacing between IFT88 and FBF1. **k**, **l** The radial distribution of IFT88 was reduced in $FBF1^{-/-}$ cells. **m** Model showing a functional role of FBF1 in maintaining the DAM spatial arrangement. Bars = 200 nm for all, except **f** (100 nm)

**DAB proteins but not FBF1 are essential for ciliogenesis.** We then examined whether DABs and the DAM have distinct functions in ciliogenesis. Consistently, depletion of the DAB component CEP164, as demonstrated previously[35,36], or SCLT1, as demonstrated herein, fully abolished DABs and ciliogenesis (Fig. 7a, b and Supplementary Fig. 16). Specifically, this resulted in defective initiation (Fig. 7a), as neither CP110 removal nor TTBK2 recruitment could occur at the mother centriole in *SCLT1*$^{-/-}$ cells (CRISPR knockout; Supplementary Fig. 14). However, intriguingly, knockout of the DAM component FBF1 did not affect ciliogenesis during the same initiation steps. Both CP110 removal and TTBK2 recruitment occurred normally in *FBF1*$^{-/-}$ cells (~ 70%), compared with WT cells (~ 80%) and yet cilia were detected in only ~ 14% of the population (Fig. 7a and Supplementary Fig. 16). These results suggest that unlike CEP164 or SCLT1, which are essential for cilia initiation, FBF1 acts downstream of SCLT1[5] and is involved in cilia maturation and/or maintenance.

**FBF1 is associated with ciliary gating of Smo and SSTR3.** To explore how FBF1 contributes to cilia maintenance, we examined whether it is required for ciliary gating. No significant difference in ciliary enrichment of ARL13B was detected between WT and *FBF1*$^{-/-}$ cilia (Fig. 7c). However, in striking contrast, although overexpressed Smoothened was highly concentrated in WT cilia, as reported previously[37], no such enrichment was observed in *FBF1*$^{-/-}$ cilia (Fig. 7d, e). Consistently, the lack of ciliary enrichment was also seen in *siFBF1* cells overexpressing SSTR3, another transmembrane protein known to localize to cilia (Fig. 7f, g). These results suggest that either the ciliary entry or retention of transmembrane proteins (Smoothened or SSTR3) is defective in *FBF1*$^{-/-}$ cilia. We repeated the assay in *CEP128*$^{-/-}$;*C-Nap1*$^{-/-}$ cells known to grow fully surfaced cilia[38], allowing clear distinction between the ciliary membrane and the plasma membrane. In the presence of FBF1, exogenously expressed Smoothened was highly enriched in surfaced cilia, but when FBF1 was depleted, as was seen in *CEP128*$^{-/-}$; *C-Nap1*$^{-/-}$;*siFBF1* cells, the level of Smoothened at surfaced cilia was much lower, although still detectable (Fig. 7h, i). If this was due to an effect on the entry of transmembrane proteins, Smo or SSTR3 should have been accumulated close to the DAPs. Therefore, our findings indicate that, in the absence of FBF1, Smoothened could enter but

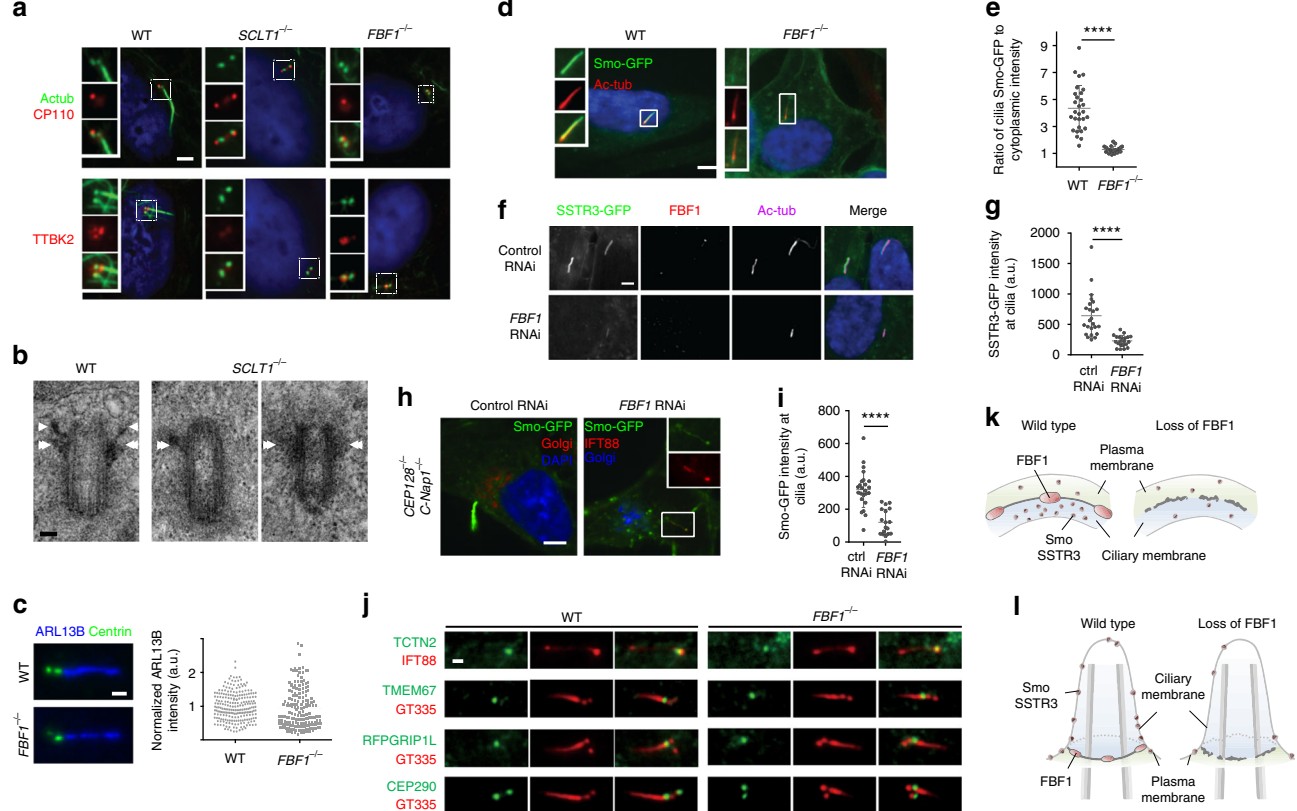

**Fig. 7** SCLT1 in ciliogenesis vs. FBF1 in transmembrane protein gating. Deletion of SCLT1 fully inhibits ciliogenesis during the initiation stage, as neither CP110 removal nor TTBK2 recruitment occurred at the mother centriole. However, both CP110 removal and TTBK2 recruitment were observed in *FBF1*$^{-/-}$ cells. White boxes highlight the areas at the basal bodies or centrioles shown in the insets. **b** EM micrographs showing the absence of DABs in *SCLT1*$^{-/-}$ cells and visible DABs in wild-type cells (arrows). Sub-distal appendages (double arrows) remain present in *SCLT1*$^{-/-}$ cells. **c** ARL13B is enriched in both WT and *FBF1*$^{-/-}$ cilia. (*n* = 227 and 170 cilia for WT and *FBF1*$^{-/-}$, respectively). **d, e** Overexpressed Smoothened was highly concentrated in WT cilia but was not enriched in *FBF1*$^{-/-}$ cilia. The insets in **d** show enlarged images of the boxed region. **e** Smo-GFP intensities relative to cytoplasmic levels. (*n* = 29 and 22 cilia for WT and *FBF1*$^{-/-}$, respectively). **f, g** SSTR3-GFP signals at cilia were lower in *siFBF1* cells. (*n* = 24 and 21 cilia for ctrl RNAi and *FBF1* RNAi, respectively in **g**). **h, i** Overexpressed Smoothened is abundant in surfaced cilia in *CEP128*$^{-/-}$;*C-Nap1*$^{-/-}$ cells, but less enriched in surfaced cilia in *CEP128*$^{-/-}$;*C-Nap1*$^{-/-}$;*siFBF1* cells, suggesting Smoothened was not retained in FBF1-depleted cilia. (*n* = 24 and 18 cilia for ctrl RNAi and *FBF1* RNAi, respectively in **i**). **j** Enrichment of transition zone proteins TCTN2, TMEM67, RPGRIP1L, and CEP290 was unaffected in *FBF1*$^{-/-}$ cells compared with WT cells. **k, l** Model showing that FBF1 plays a role in gating transmembrane proteins, presumably at the junction between the ciliary membrane and the plasma membrane. Data for **e**, **g** and **i** are presented as mean ± s.d. ****$P < 0.0001$, two-tailed Student's *t*-test. Bars = 5 μm in **a**, **d**, **f**, **h**, 100 nm in **b**, and 1 μm in **c**, **j**.

was not retained or enriched in the ciliary membrane. A possible reason for the lack of Smoothened ciliary enrichment could be a defective TZ, as a previous study showed that depletion of certain TZ proteins can inhibit this process[39]. However, we excluded this possibility by showing that TZ proteins TCTN2, TMEM67, RPGRIP1L, and CEP290 were unaffected in $FBF1^{-/-}$ cells compared with WT cells, and remained localized at the ciliary base (Fig. 7j). Our results thus strongly suggest that the DAM component FBF1 performs an essential role in ciliary gating, particularly of transmembrane proteins (Fig. 7k, l).

## Discussion

In summary, using super-resolution microscopy, we determined the detailed molecular architecture of centriolar DAPs and the surrounding structure at the ciliary base, thereby bridging the gap between structural biology and cell biology, which is essential for precise functional studies. Our work provides an unprecedented roadmap to functional investigation based on more than 12 molecular elements at the interfaces of the centriole and cilium, and the plasma and ciliary membranes. Most importantly, instead of a pinwheel-like structure based on previous EM studies, our findings reveal that DAPs have a conical-shaped architecture composed of nine DABs within an embedded matrix that we termed the DAM. Both the DABs and the DAM have distinct molecular elements that serve distinct functions. DABs contain CEP164, SCLT1, CEP89, and CEP83, all of which have an essential role in cilia initiation, whereas the previously unexplored DAM is filled with IFT molecules and FBF1, which is intriguingly found to facilitate ciliary gating of transmembrane proteins. As FBF1 is known to be associated with the apical junction complex that forms the barrier of epithelia[40], it may also have a similar role at the junction between the ciliary membrane and the plasma membrane at the base of the ciliary pockets to gate the ciliary compartment.

Accumulation of IFT88 molecules in the gaps between DABs indicates a unique molecular arrangement in this region. In order to have protein molecules concentrated/localized at a specific region of the cell, where they can dynamically enter/attach and exit/detach, the region must either be gated with a structural barrier, or alternatively include a scaffold that can bind to these protein molecules with the required affinity. In either case, dedicated structures must be present in this. In our current study, we found that FBF1 and IFT88 were concentrated/localized at the space between DABs, outside of the ciliary compartment, in a region previously thought to be empty based on EM studies. We thus conclude that, rather than being empty, the region is equipped with structures that are able to trap protein molecules to form the DAM. We use the term "matrix" to indicate a surrounding structure or material in which something develops, rather than pointing to any defined architecture.

This current work provides a functional extension of our previous finding of a hierarchical relationship among SCLT1, CEP164, and FBF1[5]. Although SCLT1 mediates the recruitment of both FBF1 and CEP164 to centrioles, recruitment of each is independent of the other. Thus, loss of FBF1 alone is completely different from loss of SCLT1, which leads to removal of both FBF1 and CEP164 from centrioles. This previously determined 'nonlinear' assembly hierarchy is consistent with the observations in our current study; both SCLT1 and CEP164 are components of DABs, and knockout of either prevents cilia initiation. By contrast, FBF1 is a matrix-associated protein occupying the space between the blades, and loss of FBF1 affects cilia maturation/maintenance (the integrity of the ciliary membrane), rather than cilia initiation.

Our previous genetic studies showed that SCLT1 acts upstream of CEP164, whereas our super-resolution studies showed that SCLT1 is "sandwiched" by the signal from the "blade-like" CEP164, not immediately obvious how this localization relationship occurs. The initial recruitment of CEP164, which depends on CEP83 and SCLT1, may be completely separate from the propagation of CEP164 along the structure that forms the final localization structure. In particular, CEP164 harbors the coiled-coil domains known to mediate self-oligomerization. That is, in the absence of CEP83/SCLT1, CEP164 molecules were not detected at centrioles. However, once initial CEP164 molecules are recruited, they can recruit other CEP164 molecules through protein-protein interaction or self-oligomerization processes. A similar concept can be seen in the PCM organization surrounding the centriole; only the inner layer of molecules is directly recruited by the centriole, whereas molecules in the outer layer are recruited through self-assembly or self-oligomerization[41]. Our previous determination of the assembly hierarchy was based on genetics, rather than the actual, stepwise, biochemical assembly pathway. The findings of the present study suggest that the basic framework of the DAP structure can in principle be assembled in the cytoplasm or in the DAP region initiated by recruited seed molecules to form the final molecular architecture.

## Methods

**Antibodies.** Detailed information about the primary antibodies against CEP164, FBF1, SCLT1, CEP89, CEP83, Cby1, CP110, CEP97, CEP162, CEP290, C2CD3, IFT88, TTBK2, ODF2, EHD1, ARL13B, and Centrin is provided in Supplementary Table 3. Antibodies conjugated to Alexa Fluor 488 (mouse A21202 and rabbit A21206; Thermo Fisher Scientific, Waltham, MA, USA) and Alexa Fluor 647 (anti-mouse A21236, anti-rabbit A21245, anti-rat A21247, and anti-goat A21447; Thermo Fisher Scientific) were used as secondary antibodies. Cy3B-conjugated secondary antibody was custom-made by conjugating Cy3B maleimide (PA63131, GE, Pittsburgh, PA, USA) to IgG antibodies (rabbit 711-005-152 and rat 712-005-153; Jackson ImmunoResearch, West Grove, PA, USA).

**Cell culture and transfection.** Human RPE cells (hTERT RPE-1, ATCC-CRL-4000) were seeded on poly-L-lysine-coated coverslips and cultured at 37 °C under 5% CO$_2$ in Dulbecco's modified Eagle's medium (DMEM)/F-12 mixture medium (1:1; 11330-032, Gibco, Thermo Fisher Scientific) supplemented with 10% fetal bovine serum (FBS) and 1% penicillin–streptomycin. HEK293T cells were cultured in DMEM with 10% FBS and 1% penicillin–streptomycin. Cilium formation in RPE-1 cells was induced by serum starvation for 24–48 h. Lentiviral stocks were prepared according to standard procedures (PT-5144-1, Clontech, Mountain View, CA, USA). RNA interference (RNAi)-mediated knockdown of human FBF1 in RPE-1 cells was performed using 12.5 μM of three different small interfering RNA (siRNA) duplex oligonucleotides (Thermo Fisher, ambion silencer select) targeting the sequences 5′-GGGAGAACCAGGCTCCAAATT-3′, 5′-GCATGGGATGCTGA-TATCTTTT-3′, and 5′-TGAGAAGCCTGTTAAACTATT-3′. siRNA oligos were delivered using Lipofectamine RNAiMAX Reagent (13778150, Thermo Fisher). Human CEP89 (NM_032816) depletion was carried out using a combination of five different lentiviral short hairpin RNAs (RNAi core, Academia Sinica, Taiwan) with the following targeting sequences: sh1, 5′-GCACGTCAAAGATATACAGAA-3′; sh 2, 5′-CAGGAGTCATTTCAAACACAT-3′; sh 3, 5′-GAATGTA-GAACTCAGTCGATA-3′; sh 4, 5′-CCAGATATAACTGGTAGAGCA-3′, and sh 5, 5′-GTAACGTTTCCGATTGTGAAA-3′. All lentiviruses were generated by transient cotransfection of HEK293T cells with packaging and envelope vectors (PT-5144-1, Clontech) using a TransIT-293 Transfection Reagent Kit (MIR 2700, Mirus, Madison, WI, USA). Stable expression of human CEP89 in RPE-1 cells was performed through lentiviral transduction and selection using 5 mg/mL puromycin. To induce ciliary growth, RPE-1 was starved of serum for 48 h. SSTR3-GFP (35623, Addgene, Cambridge, MA, USA[42]) was cloned into the pLVX-Tight-Puro vector (Clontech) for tet-inducible expression, as was Smoothened-GFP (described previously[38]). For Smo-GFP/SSRT3-GFP, lentivirus production and infection were performed as described previously[5] using psPax2 (packaging, 12260, Addgene), pCMV-VSV-G (envelope, 8454, Addgene), and pLVXTet-On Advance (Clontech) plasmids. Cells were infected simultaneously with lentiviral vectors for rttA expression (Tet-On advance) and Smo-GFP (or SSTR3-GFP). Clonal populations displaying uniform transgene expression were isolated prior to siRNA experiments. Tet induction was performed with 1 μg/mL doxycycline for 48 h (during serum starvation). Cell lines used in this study were tested for absence of mycoplasma contamination using immunofluorescence with DAPI (4′,6-diamidino-2-phenylindole) staining and a PCR assay (EZ-PCR Mycoplasma Test Kit, Biological Industries).

**CRISPR construction of *SCLT1*⁻/⁻ and *FBF1*⁻/⁻ cells.** RNA-guided targeting of genes in human cells was achieved through coexpression of the Cas9 protein with guide RNAs (gRNAs) using reagents prepared by the Church research group[43], which are available from Addgene (http://www.addgene.org/crispr/church/). Targeting sequences for SCLT1 and FBF1 were 5′-GGTGTCTGCCCATGGAAGAG-3′ and 5′-GGCATGGATGCTGATATCTT-3′, respectively, and these were cloned into the gRNA cloning vector (41824, Addgene) via the Gibson assembly method (New England Biolabs, Ipswich, MA, USA) as described previously[43]. Pure SCLT1 and FBF1 knockout cell lines were then established through clonal propagation from single cells. For genotyping, the following PCR primers were used: 5′-ACGCGGATCCAACGTAAGTGTTAGAATGCT-3′ and 5′-CTAGTCTA-GAAGCTAGGTGCATTTAACAAT-3′ for SCLT1 alleles, and 5′-ACGCG-GATCCCCCGAGACCAGCCACCTAGG-3′ and 5′-CTAGTCTAGACCTTTCTCACTGGCTACTGA-3′ for FBF1 alleles. PCR products were cloned and sequenced.

**Immunofluorescence.** Cells on glass coverslips were fixed with either 4% paraformaldehyde at room temperature or methanol at −20 °C after one wash with phosphate-buffered saline (PBS). Cells were then permeabilized with 0.1% PBST (PBS with 0.1% Triton X-100) for 10 min and blocked with 3% bovine serum albumin (BSA) for 30 min. Primary antibodies were diluted to optimized concentrations with 0.1% BSA in PBST and applied to cells. After a 1 h incubation with antibody, cells were washed three times with 0.1% PBST to remove unbound antibodies, then incubated with optimally diluted secondary antibodies for 1 h. Cells were finally washed three times with 0.1% PBST and stored in PBS with sodium azide at 4 °C until later use.

**dSTORM imaging and analysis.** Super-resolution imaging was performed with a modified inverted microscope (Eclipse Ti-E, Nikon, Tokyo, Japan). Three light sources were merged onto a laser merge module (ILE, Spectral Applied Research, Richmond Hill, Ontario, Canada), which was integrated with individual controllers. Beams from a 637 nm laser (OBIS 637 LX 140 mW, Coherent, Santa Clara, CA, USA), a 561 nm laser (Jive 561 150 mW, Cobolt, Solna, Sweden), and a 405 nm laser (OBIS 405 LX 100 mW, Coherent) were homogenized (Borealis Conditioning Unit, Spectral Applied Research) and focused on the back aperture of a 100× 1.49 numerical aperture oil-immersion objective (CFI Apo TIRF, Nikon) for wide-field illumination of samples. A perfect focusing system (Nikon) was used to minimize z-axis drift. The 637 and 561 nm laser lines were operated at a high intensity of ∼ 1−5 kW cm⁻² to quench most of the fluorescence from Alexa 647 and Cy3B. A weak 405 nm beam was used to activate a portion of the dyes that were converted from a ground state to an excited state. The collected fluorescence was cleaned by a single-band emission filter and imaged on an electron-multiplying charge-coupled device (EMCCD) camera (Evolve 512 Delta, Photometrics, Tucson, AZ, USA) with an overall pixel size of 93 nm. Fiducial markers (Tetraspeck, Thermo Fisher) were used to correct the drift. Typically, 10,000−20,000 frames (for two-dimensional dSTORM) and ∼ 50,000 frames (for 3D dSTORM) were recorded at a rate of 50 −67 fps. Individual single-molecule peaks were then real-time localized using a MetaMorph Super-resolution Module (Molecular Devices, Sunnyvale, CA, USA), based on a wavelet segmentation algorithm. Super-resolution images in figures were cleaned with a Gaussian filter with a radius of 0.6−1 pixels. For single-color imaging of Alexa 647, signals were filtered with a bandpass filter (700/75, Chroma, Bellows Falls, VT, USA); for two-color imaging, the Alexa 647 channel was first recorded, then the Cy3B channel was acquired with the corresponding filter (593/40, Chroma). For 3D imaging, a cylindrical lens with a focal length of 25 cm was added before the camera to create astigmatism, from which the z position was determined through calibration. The point spread function (PSF)-based calibration curve was generated by scanning a fluorescent bead in the z-direction and calculating the elliptical widths ($w_x$ and $w_y$).

dSTORM imaging was performed in an imaging buffer containing Tris-HCl, NaCl (TN) buffer at pH 8.0, and an oxygen-scavenging system consisting of 60 −100 mM mercaptoethylamine (MEA) at pH 8.0, 0.5 mg mL⁻¹ glucose oxidase, 40 μg mL⁻¹ catalase, and 10% glucose (Sigma-Aldrich, St. Louis, MO, USA). Lateral position drift was measured during the acquisition and corrected by frame-by-frame correlation of the fiducial markers with ImageJ. Chromatic aberration between long and short wavelength channels was compensated with a customized algorithm relocating each pixel of a 561 nm image to its targeted position in the 647 nm channel with a predefined correction function obtained by parabolic mapping of multiple calibration beads. The system resolutions for Alexa 647 and Cy3B were characterized with fixed samples incubated in an imaging buffer of 80 mM MEA and evaluated via the analysis of localization precision as a function of the photon count, the noise level, the pixel size of a camera, and the width of a PSF of single-molecule emission (Supplementary Fig. 1)[44]. For axial imaging of DAPs, a ring pattern of labeled SCLT1 or FBF1 was usually sought to identify those with appropriate orientations. For lateral imaging, a rod-like pattern of SCLT1 or FBF1 was used to assure a lateral view of the centriole−cilium.

To determine the diameter of DAPs and their associated proteins shown in Fig. 1b, the position of individual super-resolved puncta forming a ring-shaped pattern was first measured and then fitted with a circle to estimate its diameter. For CEP164, Cby1, IFT88, and TTBK2, where their distributions were relatively dispersed, their radial positions were described with a radius defined as the distance between the puncta and the center, yielding a radius histogram that was fitted with a Gaussian or mixed Gaussian curve (Supplementary Fig. 2). To determine how DAP proteins aligned with each other (Fig. 2a−e), angular analysis was performed. Briefly, the angular position of each DAP punctum was recorded, and angle differences between each pair of puncta for two protein species indicated how they were arranged relative to each other. The concept is described in detailed in Supplementary Fig. 5. To align dSTORM and EM images in Fig. 3d, FBF1 signals of dSTORM images were placed at the tip of the high electron density signals of slanted DAPs in EM images.

**Transmission electron microscopy.** RPE-1 cells grown on coverslips made of Aclar film (Electron Microscopy Sciences, Hatfield, PA, USA) were fixed in 4% paraformaldehyde and 2.5% glutaraldehyde with 0.1% tannic acid in 0.1 M sodium cacodylate buffer at room temperature for 30 min, postfixed in 1% OsO₄ in sodium cacodylate buffer for 30 min on ice, dehydrated in a graded series of ethanol, infiltrated with EPON812 resin (Electron Microscopy Sciences), and embedded in the resin. Serial sections (∼ 90 nm thickness) were cut on a microtome (Ultracut UC6; Leica, Wetzlar, Germany) and stained with 1% uranyl acetate and 1% lead citrate. Samples were examined on a transmission electron microscope (Tecnai G2Spirit Twin, FEI, Hillsboro, OR, USA).

**Three-dimensional printing model.** To reconstruct a 3D model of DAPs, we first drew the DAP structure using 3D CAD design software (SOLIDWORKS, Waltham, MA, USA) then fabricated a solid model with a 3D printer (Form 1 +, Formlabs, Somerville, MA, USA). The dimensions of the model were based on the localization positions obtained from the super-resolution results in Fig. 1b, d, Fig. 2a-f, and Fig. 3a, c, e, f, and on information from previous studies[7,10]. For the 3D printed model, DAP proteins in each "blade" were printed as spherical shapes with their centers at the mean positions, then painted in the appropriate colors. The printed model, including microtubule triplets and doublets, DABs, TZ, and ciliary pocket, has an overall size of 85 mm in width (diameter) by 155 mm in height at a 150,000:1 scale. To properly place FBF1 in the gaps between two adjacent DAPs, an auxiliary annular scaffold, which does not exist in the actual complex, was added.

**Data availability.** The authors declare that all data supporting the findings of this study are available within the article and its Supplementary Information files or from the corresponding author upon reasonable request.

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

# ARTICLE

15. Lu, Q. et al. Early steps in primary cilium assembly require EHD1/EHD3-dependent ciliary vesicle formation. *Nat. Cell Biol.* **17**, 228–240 (2015).

16. Caspary, T., Larkins, C. E. & Anderson, K. V. The graded response to Sonic Hedgehog depends on cilia architecture. *Dev. Cell* **12**, 767–778 (2007).

17. Deane, J. A., Cole, D. G., Seeley, E. S., Diener, D. R. & Rosenbaum, J. L. Localization of intraflagellar transport protein IFT52 identifies basal body transitional fibers as the docking site for IFT particles. *Curr. Biol.* **11**, 1586–1590 (2001).

18. Spektor, A., Tsang, W. Y., Khoo, D. & Dynlacht, B. D. Cep97 and CP110 suppress a cilia assembly program. *Cell* **130**, 678–690 (2007).

19. Reiter, J. F., Blacque, O. E. & Leroux, M. R. The base of the cilium: roles for transition fibres and the transition zone in ciliary formation, maintenance and compartmentalization. *EMBO Rep.* **13**, 608–618 (2012).

20. Chang, B. et al. In-frame deletion in a novel centrosomal/ciliary protein CEP290/NPHP6 perturbs its interaction with RPGR and results in early-onset retinal degeneration in the rd16 mouse. *Hum. Mol. Genet.* **15**, 1847–1857 (2006).

21. Wang, W.-J. et al. CEP162 is an axoneme-recognition protein promoting ciliary transition zone assembly at the cilia base. *Nat. Cell Biol.* **15**, 591–601 (2013).

22. Rosenbaum, J. L. & Witman, G. B. Intraflagellar transport. *Nat. Rev. Mol. Cell Biol.* **3**, 813–825 (2002).

23. Rust, M. J., Bates, M. & Zhuang, X. W. Sub-diffraction-limit imaging by stochastic optical reconstruction microscopy (STORM). *Nat. Methods* **3**, 793–795 (2006).

24. Heilemann, M. et al. Subdiffraction-resolution fluorescence imaging with conventional fluorescent probes. *Angew. Chem. Int. Ed.* **47**, 6172–6176 (2008).

25. Yang, T. T. et al. Superresolution pattern recognition reveals the architectural map of the ciliary transition zone. *Sci. Rep.* **5**, 14096 (2015).

26. Lau, L., Lee, Y. L., Sahl, S. J., Stearns, T. & Moerner, W. E. STED microscopy with optimized labeling density reveals 9-fold arrangement of a centriole protein. *Biophys. J.* **102**, 2926–2935 (2012).

27. Lee, Y. L. et al. Cby1 promotes Ahi1 recruitment to a ring-shaped domain at the centriole–cilium interface and facilitates proper cilium formation and function. *Mol. Biol. Cell* **25**, 2919–2933 (2014).

28. Sillibourne, J. E. et al. Assessing the localization of centrosomal proteins by PALM/STORM nanoscopy. *Cytoskeleton* **68**, 619–627 (2011).

29. Sonnen, K. F., Schermelleh, L., Leonhardt, H. & Nigg, E. A. 3D-structured illumination microscopy provides novel insight into architecture of human centrosomes. *Biol. Open* **1**, 965–976 (2012).

30. Cevik, S. et al. Joubert syndrome Arl13b functions at ciliary membranes and stabilizes protein transport in Caenorhabditis elegans. *J. Cell Biol.* **188**, 953–969 (2010).

31. Joo, K. et al. CCDC41 is required for ciliary vesicle docking to the mother centriole. *Proc. Natl Acad. Sci. USA* **110**, 5987–5992 (2013).

32. Jensen, V. L. et al. Formation of the transition zone by Mks5/Rpgrip1L establishes a ciliary zone of exclusion (CIZE) that compartmentalises ciliary signalling proteins and controls PIP2 ciliary abundance. *EMBO J.* **34**, 2537–2556 (2015).

33. Cajanek, L. & Nigg, E. A. Cep164 triggers ciliogenesis by recruiting Tau tubulin kinase 2 to the mother centriole. *Proc. Natl Acad. Sci. USA* **111**, E2841–E2850 (2014).

34. Wei, Q. et al. Transition fibre protein FBF1 is required for the ciliary entry of assembled intraflagellar transport complexes. *Nat. Commun.* **4**, 2750 (2013).

35. Graser, S. et al. Cep164, a novel centriole appendage protein required for primary cilium formation. *J. Cell Biol.* **179**, 321–330 (2007).

36. Schmidt, K. N. et al. Cep164 mediates vesicular docking to the mother centriole during early steps of ciliogenesis. *J. Cell Biol.* **199**, 1083–1101 (2012).

37. Corbit, K. C., Aanstad, P., Singla, V. & Norman, A. R. Vertebrate Smoothened functions at the primary cilium. *Nature* **437**, 1018 (2005).

38. Mazo, G., Soplop, N., Wang, W.-J., Uryu, K. & Tsou, M.-F. B. Spatial control of primary ciliogenesis by subdistal appendages alters sensation-associated properties of cilia. *Dev. Cell* **39**, 424–437 (2016).

39. Garcia-Gonzalo, F. et al. A transition zone complex regulates mammalian ciliogenesis and ciliary membrane composition. *Nat. Genet* **43**, 776–784 (2011).

40. Sugimoto, M. et al. The keratin-binding protein Albatross regulates polarization of epithelial cells. *J. Cell Biol.* **183**, 19–28 (2008).

41. Mennella, V. et al. Subdiffraction-resolution fluorescence microscopy reveals a domain of the centrosome critical for pericentriolar material organization. *Nat. Cell Biol.* **14**, 1159–1168 (2012).

42. Berbari, N., Johnson, A., Lewis, J., Askwith, C. & Mykytyn, K. Identification of ciliary localization sequences within the third intracellular loop of G protein-coupled receptors. *Mol. Biol. Cell* **19**, 1540–1547 (2008).

43. Mali, P. et al. RNA-guided human genome engineering via Cas9. *Science* **339**, 823–826 (2013).

44. Yildiz, A. et al. Myosin V walks hand-over-hand: single fluorophore imaging with 1.5-nm localization. *Science* **300**, 2061–2065 (2003).

## Acknowledgements

We thank Erich Nigg, Christopher Westlake, Steve Caplan, and Tang Tang for sharing reagents. This work was supported by the Ministry of Science and Technology, Taiwan (Grant Number 103-2112-M-001-039-MY3), Academia Sinica Career Development Award, Academia Sinica Nano Program, the University of Alabama at Birmingham (UAB) Hepato/Renal Fibrocystic Diseases Core Center (HRFDCC) Pilot Award (NIH 5P30DK074038-09) to J.-C.L.; NIH grant GM088253, American Cancer Society grant RSG-14-153-01-CCG, and the Geoffrey Beene Cancer Research Center grant to M.-F.B. T.

## Author contributions

J.-C.L., M.-F.B.T., and T.T.Y. conceived the study. T.T.Y. performed the super-resolution imaging and analyzed the data. W.M.C. performed the super-resolution imaging. W.-J. W. performed the CRISPR and TEM works. G.M. performed the lentiviral work and the stable line generation. B.T. created the CRISPR lines. Z.C. created the 3D computational model. T.M.N.T. assisted the lentiviral work. Y.-D.C. assisted the super-resolution imaging. R.R.W. assisted the FBF1 study. C.-E.H. assisted the siRNA depletion work. W.-N.J. performed the TEM work. T.T.Y., J.-C.L., and M.-F.B.T. interpreted the data and wrote the manuscript.

## Additional information

**Competing interests:** The authors declare no competing interests.

