## [Peer Review File · Nature Communications]

Reviewers' comments:

Reviewer #1 (Remarks to the Author):

The authors have done a good job at addressing all my comments. I have no further comments.

Reviewer #2 (Remarks to the Author):

I was one of the original reviewers of this manuscript and was supportive of publication with minor revisions in the first instance. In the revised version of their manuscript, the authors have done a fantastic job addressing most if not all of my concerns (and those of the other reviewers). This is an excellent study that will be of much interest to the broad readership of Nature Communications.

Reviewer #3 (Remarks to the Author):

I can only comment on the technical aspects of the STORM imaging.

I share the concerns of reviewer 3 with regard to the lack of evaluation of the performance of the STORM imaging system, and in particular as it relates to Figure 2f. Firstly, Reviewer 3 quite reasonably asked for some evaluation of the resolution of the system; this has not been done. It is not good enough to state the value is 'about 20nm' in the response. What is the resolution in the datasets they have? At the very least, they should give a distribution of the photon numbers per molecule and the average background level, and calculate the best case scenario precision based resolution from that (though it should be noted that this is a best case figure, and the actual value may be lower). If they have carried out calibrations, these should also be described. What test samples were used? How was the resolution evaluated?

Part of the reason I am so concerned about the evaluation is Figure 3f. Why do the error bars vary so much, particularly whether they are larger along the axial or radius axis? Is this due to the much larger uncertainty in z than in xy? I find it hard to imagine what led to this, which I think means it clearly needs some more explanation.

Having said all that, I do think the results are very impressive and the data is very good quality. But even good quality data needs to be evaluated rigorously and clearly so that everyone knows how the results were arrived at.

1. I share the concerns of reviewer 3 with regard to the lack of evaluation of the performance of the STORM imaging system, and in particular as it relates to Figure 2f. Firstly, Reviewer 3 quite reasonably asked for some evaluation of the resolution of the system; this has not been done. It is not good enough to state the value is 'about 20nm' in the response. What is the resolution in the datasets they have? At the very least, they should give a distribution of the photon numbers per molecule and the average background level, and calculate the best case scenario precision based resolution from that (though it should be noted that this is a best case figure, and the actual value may be lower). If they have carried out calibrations, these should also be described. What test samples were used? How was the resolution evaluated?

A: We would like to thank the reviewer for this constructive comment. To address the reviewer's concern on the lack of a clear message of the resolution evaluation, we have provided the measured photon distributions of Alexa 647 and Cy3B in Figure R1A where samples were fixed with methanol and labeled with typical ciliary markers. The samples incubated in an imaging buffer containing 80 mM MEA were illuminated at an intensity of around 2 kW/cm². The number of photons detected per switching event was registered as the total integrated signal minus the average background level and then converted to photon counts using the electron multiplication gain settings and analog-to-digital conversion gain (values shown in Figure R1B). The localization precision (σ) of our system was estimated on the basis of the equation¹

$$\sigma = \sqrt{\frac{s^2}{N} + \frac{a^2/12}{N} + \frac{8\pi s^4 b^2}{a^2 N^2}}$$

and the associated parameters were listed in Figure R1B. The standard deviation (s) of a PSF was obtained through a 2D Gaussian function-fit to the PSF distribution. The mean values from around ten molecules for Alexa 647 and Cy3B were 148 nm and 134 nm, respectively. For the photon count (N), we used the value at the peak occurrence frequency of the photon number distribution for each case (Figure R1A). To characterize the average background DC level and the standard deviation of the background (b), intensity histograms of three image sets with noises only were analyzed and the values were converted to effective photon counts. The localization precisions in FWHM were thus summarized for the best case scenario in Figure R1B, showing a precision of 18.76 nm and 24.65 nm for Alexa 647 and Cy3B, respectively. We have mentioned the estimation of precision localization in the main text (page 5) and have added Supplementary Fig. 1 (the same as Figure R1). We have also added the detail of how we estimated our precision localization in the Methods section (page 35).

Figure R1 Evaluation of localization precision of the dSTORM system. (A) Measured photon distributions of Alexa 647 and Cy3B from fixed samples. (B) Estimation of localization uncertainty in the xy plane showing the system resolution of ~19 and ~25 nm for Alexa 647 and Cy3B, respectively.

2. Part of the reason I am so concerned about the evaluation is Figure 3f. Why do the error bars vary so much, particularly whether they are larger along the axial or radius axis? Is this due to the much larger uncertainty in z than in xy? I find it hard to imagine what led to this, which I think means it clearly needs some more explanation.

Having said all that, I do think the results are very impressive and the data is very good quality. But even good quality data needs to be evaluated rigorously and clearly so that everyone knows how the results were arrived at.

A: First, we would like to thank the reviewer for appreciating our superresolution imaging studies of the distal appendage architecture. The following is to clarify the reviewer's major concern on the different error bars for different proteins in Figure 3f. The reviewer actually pointed out an important finding of this study regarding the

distinct localizations and varying distributions of several DAP proteins which were depicted with their mean positions and standard deviations (error bars) in Fig. 3f, respectively. As described in the original manuscript, in fact, the axial and radial results were all collected by two-dimensional dSTORM imaging but under different imaging orientations of the centrioles, i.e. a top-view observation for the radial measurements and a side-view observation for the axial measurements. Therefore, the axial and radial imaging analyses theoretically shared the same localization precision in xy but did not depend upon the z-axis localization uncertainty. The major factor that contributed to the varying error bars was due to the intrinsic differences in distributions among different DAP proteins. For example, CEP89 proteins of the top layer were more concisely localized and well aligned with the location of FBF1 (small standard deviation in Figure 3f), whereas CEP89 proteins of the bottom layer were distributed slightly more broadly in space (Figure R2A) (larger standard deviation in Figure 3f). A relatively large variation was also found in the axial distribution of ODF2 (Figure R2B, C) as compared with those of FBF1/SCLT1/CEP164, each of which localized to a more confined axial level (Figure R2D). A broad distribution of a protein can possibly reflect its multiple docking/binding sites or a structural/motif role that require an elongated occupancy. We have added a note in page 7 to clarify the main cause of diverse localization variations in Fig. 3f due to different spatial distribution ranges of different proteins.

Figure R2 Intrinsic differences in axial distributions of different DAP proteins. Relatively large variations of localizations were found for CEP89 (A) and ODF2 (B). (C) A histogram of the axial localization of CEP89 demonstrating broad distributions of two distinguishable layers. (D) Relatively confined axial distributions of FBF1, SCLT1, and CEP164.

Reference

1. Yildiz, A. *et al.* Myosin V walks hand-over-hand: single fluorophore imaging with 1.5-nm localization. *science* **300**, 2061-2065 (2003).